# Parthenogenetic Reproduction in *Strumigenys* Ants: An Update

**DOI:** 10.3390/insects14020195

**Published:** 2023-02-16

**Authors:** Chu Wang, Ping-Jui Sung, Chung-Chi Lin, Fuminori Ito, Johan Billen

**Affiliations:** 1Zoological Institute, KU Leuven, Naamsestraat 59, P.O. Box 2466, B-3000 Leuven, Belgium; 2Department of Biology, College of Science, National Changhua University of Education, Changhua 50007, Taiwan; 3Faculty of Agriculture, Kagawa University, Ikenobe 2393, Miki 761-0795, Japan

**Keywords:** spermatheca, spermatheca gland, thelytoky, morphology, histology, ultrastructure

## Abstract

**Simple Summary:**

Similar to wasps and bees, ants also produce their males by parthenogenesis (development from unfertilized eggs). Much more exceptional is thelytoky, which means that female offspring is also produced by parthenogenesis. This phenomenon to date has been reported for only 16 ant species, including three that belong to the genus *Strumigenys*: *S. hexamera*, *S. membranifera* and *S. rogeri*. In the present study we discovered that thelytoky also occurs in three more *Strumigenys* species: *S. emmae*, *S. liukueiensis* and *S. solifontis*. Although queens of these thelytokous *Strumigenys* species do not need sperm to produce female offspring, they all do contain a functional reproductive system. This includes the presence of a spermatheca, which is the storage organ in which queens store the sperm they receive from the males during the mating flight. We hypothesize that these queens retain a spermatheca for the rare event when they encounter a male and mate with it. In several of these thelytokous *Strumigenys* species, males are extremely rare but are occasionally produced. Mating with such males considerably increases the genetic variability of the offspring in these thelytokous *Strumigenys* colonies. Several thelytokous species, such as *S. emmae*, *S. membranifera* and *S. rogeri,* are tramp species. It is obvious that reproduction without the need to mate with males offers these species a big advantage when establishing colonies in new environments.

**Abstract:**

Parthenogenetic reproduction is a common feature for social Hymenoptera, as males typically develop from unfertilized eggs (arrhenotoky). Production of female offspring without the involvement of sperm (thelytoky) also exists but is rather exceptional as it has been reported for only 16 ant species so far. Three of these belong to the genus *Strumigenys: S. hexamera*, *S. membranifera* and *S. rogeri*. Our observations on the reproductive biology in various Oriental *Strumigenys* species extends this list of thelytokous ants with three more species: *S. emmae*, *S. liukueiensis* and *S. solifontis*. Of these six thelotykous species, *S. emmae*, *S. membranifera* and *S. rogeri* are known as tramp species. Reproduction without the need to fertilize eggs no doubt offers these species a considerable advantage when establishing colonies in new environments. Published histological data on *S. hexamera* and *S. membranifera* already showed that the queens possess a functional spermatheca. We now provide evidence that this is also the case for the four other thelytokous *Strumigenys* species. Retaining a functional spermatheca and reproductive system may keep the queens ready for the exceptional event of mating and hence increase genetic variability, as males do occur very rarely.

## 1. Introduction

Haplodiploid reproduction is one of the main characteristics in the evolution of social Hymenoptera [1]. In this system, females are diploid and originate from fertilized eggs, whereas males are haploid and develop from unfertilized eggs through arrhenotokous parthenogenesis. In several eusocial hymenopteran species, however, thelytokous parthenogenesis has been demonstrated also, in which diploid female offspring are produced by unmated females [2]. Reproduction in this case becomes possible without the need to search for mates and copulation, which represents an obvious advantage in the spreading of tramp species. Thelytoky has so far been reported for 16 ant species representing 4 subfamilies: *Ooceraea biroi* [3] in the Dorylinae, *Platythyrea punctata* [4] in the Ponerinae, *Cataglyphis cursor* [5], *C. hispanica* [6] and *Paratrechina longicornis* [7] in the Formicinae, and 10 species in the Myrmicinae: *Messor capitatus* [8], *Monomorium hiten* [9] and *M. triviale* [10], *Mycocepurus smithii* [11], *Myrmecina nipponica* [12], *Pristomyrmex punctatus* [13], *Strumigenys hexamera* [14], *S. membranifera* [15] and *S. rogeri* [16], *Vollenhovia emeryi* [17] and *Wasmannia auropunctata* [18].

As thelytokous reproduction makes males and their sperm no longer needed, the need for retaining a spermatheca can be questioned. For five of the above parthenogenetic species, however, the presence of a functional spermatheca in which sperm can be kept alive and used to fertilize oocytes has been histologically confirmed: *Monomorium triviale*, *Pristomyrmex punctatus* and *Strumigenys membranifera* [19], *S. hexamera* [14] and *Myrmecina nipponica* [12]. In *Paratrechina longicornis* queens, the presence of live, viable sperm in dissected spermathecae could be confirmed [20]. Our study of reproductive behaviour in a variety of Oriental *Strumigenys* species revealed that besides *S. hexamera*, *S. membranifera* and *S. rogeri* (the latter without histological analysis), at least another three species are thelytokous as well: *S. emmae*, *S. liukueiensis* and *S. solifontis*. We report here on our observations of these species and provide the results of a histological and ultrastructural examination of their reproductive system. 

## 2. Material and Methods

Queenright *Strumigenys* colonies were collected from the field in various sites in Taiwan (locality information in Table 1 and Table 2). They were transferred to artificial nests in the laboratory that consisted of a round plastic tray with a diameter of 21.5 cm and a floor cover made of plaster of Paris to provide moisture. The colonies were reared under a 12:12 (L:D) photoperiod at 24 ± 2 °C in an incubation room. Springtails (*Cyphoderus albinus*) were provided as food ad libitum while a glass tube with water and cotton was provided to keep sufficient humidity. The trays were covered with a plastic lid to prevent escape by the ants and the springtails. We assured that alate queens (F1 generation) were virgin by raising them in the strict absence of males in the source colonies (F0 generation). Experimental colonies were created, each consisting of 1 alate queen (F1 queen) and 15 nestmate workers. Colony development (Table 1) was recorded twice a week during an experimental period of 28 weeks to monitor whether virgin queens can lay eggs that thelytokously develop into workers and/or queens (for *S. membranifera*, the total experimental period was 202 weeks). The virgin condition of all alate queens was verified by dissecting them at the end of the experimental period, which confirmed that their spermatheca was empty. To obtain additional evidence of whether mated queens occur, we dissected field-collected dealate queens for examination of their ovaries and spermatheca contents (Table 2).

Histological and ultrastructural examination was performed on queens of the thelytokous *S. emmae*, *S. liukueiensis*, *S. rogeri* and *S. solifontis*, as well as on the non-thelytokous *S. lacunosa* (Lugu Township, Nantou County, Taiwan), *S. nanzanensis* (Lanyu Township, Taitung County, Taiwan), *S. sauteri* (Lugu Township, Nantou County, Taiwan) and *S. sydorata* (Bogor, Indonesia) for comparison. The posterior part of the gaster was cut off and fixed in 2% glutaraldehyde (Agar Scientific, Stansted, UK) buffered with 50 mM Na-cacodylate (Agar Scientific, Stansted, UK) and 150 mM saccharose (pH 7.3). Postfixation was carried out in 2% osmium tetroxide (Agar Scientific, Stansted, UK) in the same buffer and was followed by dehydration in a graded acetone series and embedding in Araldite (Polysciences, Warrington, PA, USA and Merck, Darmstadt, Germany). Longitudinal serial semithin sections of 1 µm thickness were made with a Leica EM UC6 ultramicrotome (Leica, Wetzlar, Germany). They were stained with methylene blue and thionin (Merck, Darmstadt, Germany) and examined with an Olympus BX-51 microscope (Olympus, Tokyo, Japan). Thin sections of 70 nm were double-stained with lead citrate (Merck, Darmstadt, Germany) and uranyl acetate (Polysciences, Warrington, PA, USA) and were examined with a Zeiss EM900 electron microscope (Zeiss, Oberkochen, Germany).

## 3. Results

### 3.1. Reproduction Biology

Observation and assessment of offspring production in experimental colonies with unfertilized queens in various *Strumigenys* species showed that 5 species produced worker and queen offspring and thus displayed thelytokous reproduction (Table 1). For three of these, we show the occurrence of thelytoky for the first time: *S. emmae, S. liukueiensis* and *S. solifontis.* In *S. lacunosa* and *S. minutula*, on the contrary, only male offspring appeared. Males were not produced at all (*S. emmae*, *S. liukueiensis* and *S. solifontis*) or in very low numbers (*S. membranifera* and *S. rogeri*) in the thelytokous species. Dissection data of field-collected dealate queens showed that all but one virgin queen of *S. hispida* had developed ovaries with yellow bodies, and provided additional support for the thelytokous reproduction in *S. liukueiensis*, *S. rogeri* and *S. solifontis*, as none of the examined queens in these species were inseminated (Table 2). 

During oviposition, the queen keeps the sting extruded to allow passing of the egg (F. Ito, unpublished observations).

### 3.2. Histology and Ultrastructure

The female genital tract of *Strumigenys* queens consists of 2 ovarioles at each side (although 5–10 per side in *S. mutica*: F. Ito, personal communication) that open into the ipsilateral lateral oviduct. The two mesodermal and hence not cuticle-lined lateral oviducts join together in the common cuticle-lined oviduct that is ectodermal. The distal portion of the common oviduct is differentiated into a thick-walled bursa copulatrix, while the thin-walled proximal portion of this common oviduct opens ventrally of the sting to the outside (Figure 1 and Figure 2). This proximal part of the common oviduct appears S-shaped when the sting is retracted, allowing it to be stretched during oviposition when the sting is extruded. The spermatheca opens into the dorsal wall of the anterior bursa copulatrix. It appears as an ovoid sac of which the epithelial wall is thickened at one side with cylindrical gland cells (the ‘hilar epithelium’, which occurs near the spermatheca duct), while the opposite wall is formed by a thin epithelium with flattened cells (the ‘distal epithelium’). The absolute thickness values of the thick and thin epithelium vary among species, but their ratio is generally around 3:1 (Figure 3). Semi-thin sections through the queen abdomen of *S. emmae*, *S. liukueiensis*, *S. rogeri* and *S. solifontis* confirm the presence of a spermatheca (Figure 3A–D), which is similar to that of non-thelytokous species (Figure 3E,F), except that the latter after mating contains sperm (Figure 3E).

Ultrastructural examination reveals clear differences in cytoplasmic composition between the thick and thin epithelium. Whereas the thin epithelium shows few or no organelles (Figure 4A,B), the thick epithelium contains numerous mitochondria, a differentiation of the apical cell membrane into a microvillar border and a tortuous course of the intercellular cell membranes (Figure 4C–E).

The spermatheca in its anterior part is connected to a paired spermatheca gland (Figure 3A–E). This gland belongs to class-3 following the standard classification of insect exocrine glands [21] with secretory cells that are connected by duct cells to the spermatheca gland duct (Figure 5A,C). The junction between both cells is formed by the end apparatus, which is a cuticular canal surrounded by microvilli that drains the secretion from the secretory cells to the reservoir. The long microvilli can appear in a tightly packed configuration (Figure 5A,B) or are loosely arranged with large spaces between them (Figure 5C,D). The cytoplasm of the secretory cells contains prominent accumulations of glycogen (Figure 5A,C,E), numerous mitochondria (Figure 5A,B) and a well-developed Golgi apparatus (Figure 5E).

## 4. Discussion

Our observations on the offspring production by experimental colonies with unfertilized queens showed that several *Strumigenys* species display thelytokous reproduction. For *S. membranifera* [15] and *S. rogeri* [16], this had already been reported in the literature, as it had been for *S. hexamera* [14]. Our finding that *S. emmae*, *S. liukueiensis* and *S. solifontis* are thelytokous is novel, however, and brings the number of thelytokous ant species to 19, of which 6 belong to the genus *Strumigenys*. Except for the monogynous *S. hexamera* [14], the other species are all polygynous. The six parthenogenetic species include both short- and long-mandibulate *Strumigenys*, which illustrates that this reproductive mode is not limited to a specific group. It is also worth noting that three of these thelytokous species are tramp species: *S. emmae* [22,23], *S. membranifera* [24] and *S. rogeri* [25]. It is obvious that thelytoky considerably helps tramp species in their spreading, as colonies can grow more easily when queens no longer need to search for males and mate with them [2]. Light and electron microscopy examination clearly shows that queens of *S. emmae*, *S. liukueiensis*, *S. rogeri* and *S. solifontis* have a functional female reproductive system similar to that in non-parthenogenetic species. Having ovarioles and oviducts is evident, as parthenogenetic queens also lay non-fertilized eggs. Although they do not need sperm to produce female offspring, they do possess a spermatheca with its associated glands and a bursa copulatrix, which is the organ in which ejaculates during copulation are initially deposited before the sperm is stored in the spermatheca [26,27]. The functionality of the spermatheca in being able to store sperm alive and use it to fertilize oocytes is apparent because of the presence of a thick hilar epithelium with columnar cells [28,29,30]. The cytoplasmic composition of the thick epithelium with numerous mitochondria, apical microvilli and tortuous intercellular membranes has also been reported in previous studies of the spermatheca in ants [29]. The ultrastructural features of the spermatheca glands with a prominent end apparatus, glycogen accumulations, numerous mitochondria and a well-developed Golgi apparatus are equally in line with those in earlier reports [28,29]. The hilar epithelium of the spermatheca reservoir provides the environment to keep the stored sperm viable for many years, while the secretion of the spermatheca glands may activate metabolically arrested sperm when it leaves the spermatheca prior to its descent toward the oviduct to fertilize the eggs [28]. In case of a degenerated spermatheca, microvilli and basal invaginations are absent while only few cytoplasmic organelles occur [29].

The presence of a functional spermatheca and its associated glands was also reported for *S. membranifera* [19] and *S. hexamera* [14], which means this feature is thus shared by all six known thelytokous *Strumigenys* species. Thelytokous reproduction, especially if permanent, may cause degeneration of the female’s ability to mate and therefore may also mean that a functional reproductive apparatus including a spermatheca is no longer needed. According to Gotoh et al. [19], the short time during which thelytoky has evolved may not be long enough to allow the spermatheca to degenerate. Being permanently thelytokous would lead to reduced heterozygosity with the risk of a genetic dead end, which could be avoided if occasional mating and an increase in intracolonial genetic variation would occur. The high occurrence of gynandromorphs in colonies of *Vollenhovia emeryi* due to complete absence of sexual reproduction is an example of where low genetic variability can lead [31]. Retaining a functional reproductive apparatus, therefore, can also be understood in keeping the queens ready in case of the rare, but not impossible, encounter and mating with a male. Masuko [14] reported that males may very occasionally occur in *S. hexamera*, as he found 4 males during 34 years of observation. Exceptional males have also been found in *S. membranifera* (8 males during 10 years of observation: C.-C. Lin, unpublished observations), *S. rogeri* (5 males during 10 years: C.-C. Lin, unpublished observations) and *S. solifontis* (2 males during 10 years: C.-C. Lin, unpublished observations), as well as in the present study (Table 1). Males have also been reported for *S. emmae* with a published picture (CASENT0133418) of a male head in AntWeb [32]. They have not yet been found in *S. liukueiensis*, but considering the extremely low frequency of male occurrence in the other thelytokous species, their existence cannot be excluded. Very rare occurrences of males have also been reported for other thelytokous ant species as *Pristomyrmex punctatus* [13] and *Monomorium hiten* [9], which supports the hypothesis that thelytokous species retain a functional spermatheca for the rare event of mating.

## 5. Conclusions

Our study reports on the existence of thelytokous reproduction in *Strumigenys emmae*, *S. liukueiensis* and *S. solifontis*, thus bringing the total number of thelytokous ant species to 19, of which 6 belong to the genus *Strumigenys*. The queens of these 6 species are all characterized by the presence of a functional reproductive apparatus with a bursa copulatrix, a spermatheca with thick and thin epidermis and spermatheca glands. This can be understood in their staying ‘standby’ for the rare event of encountering and mating with males, which has been reported sporadically in five of the six species.

## Figures and Tables

**Figure 1 insects-14-00195-f001:**
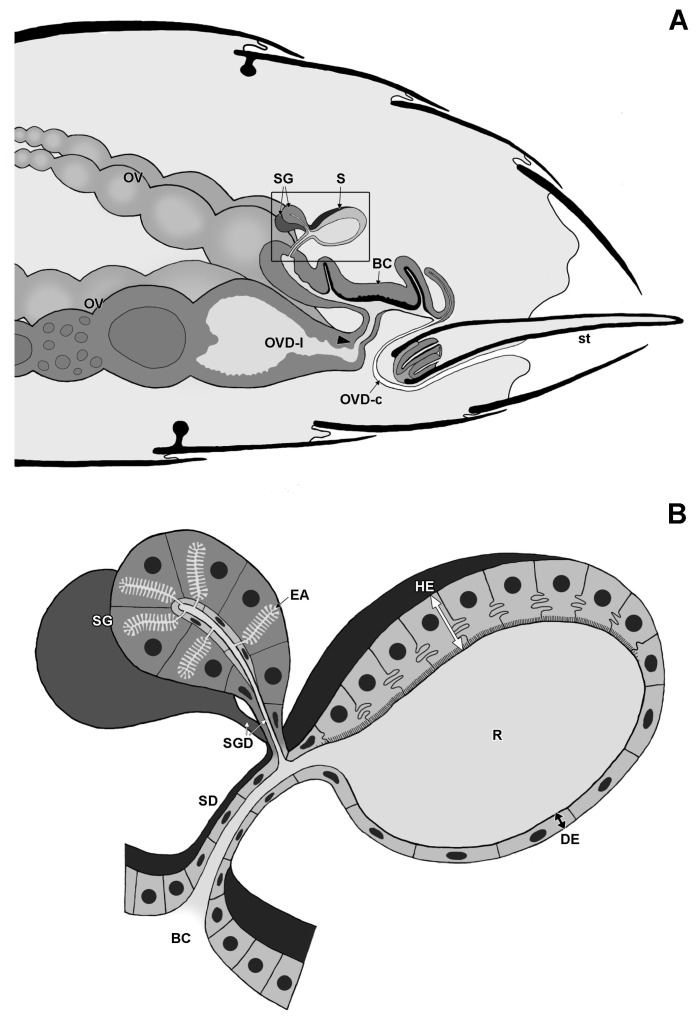
Schematic drawing of gaster with reproductive system (**A**) and spermatheca (**B**) in *Strumigenys* queens. Note S-shaped oviduct when sting is in retracted rest position. BC: bursa copulatrix, DE: distal epithelium, EA: end apparatus, HE: hilar epithelium, OV: ovariole, OVD-l: lateral oviduct, OVD-c: common oviduct, R: reservoir, S: spermatheca, SD: spermatheca duct, SG: spermatheca gland, SGD: spermatheca gland duct, st: sting. Arrowhead indicates transition between common (cuticle-lined) and lateral (no cuticular lining) oviduct.

**Figure 2 insects-14-00195-f002:**
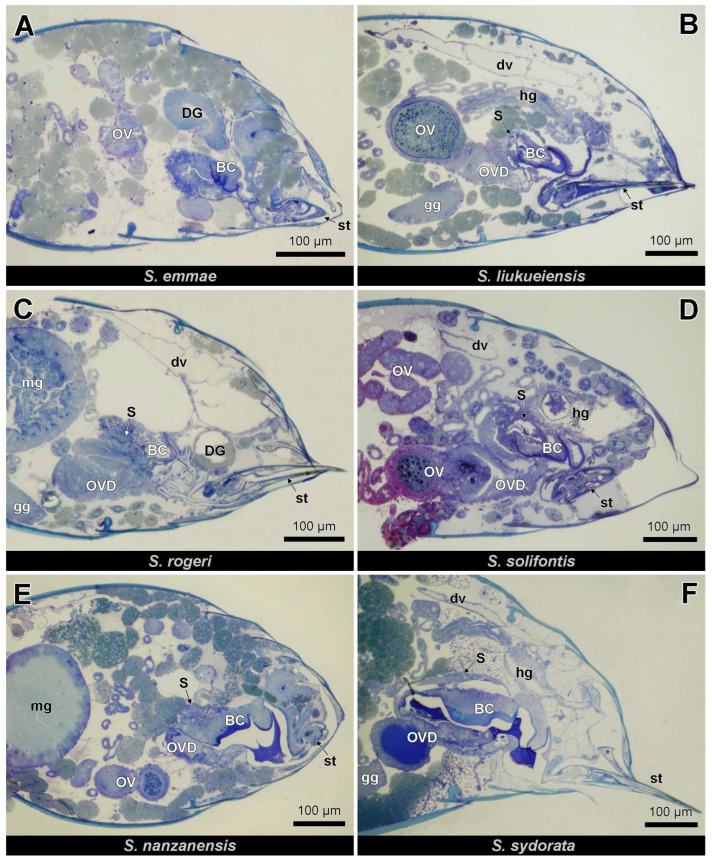
Semi-thin longitudinal sections (anterior to the left) of queen gaster in four parthenogenetic species (**A**–**D**) and two non-parthenogenetic species (**E**,**F**). BC: bursa copulatrix, DG: Dufour gland, dv: dorsal vessel, gg: ganglion, hg: hindgut, mg: midgut, OV: ovariole, OVD: oviduct, S: spermatheca, st: sting.

**Figure 3 insects-14-00195-f003:**
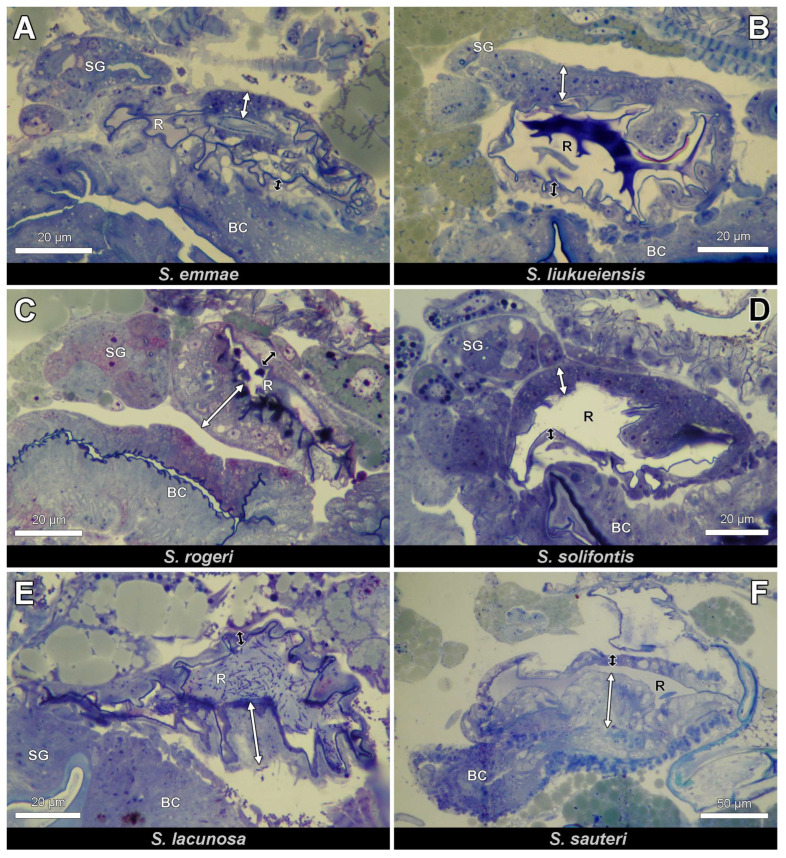
Close-up views (longitudinal sections, anterior to the left) of queen spermatheca in four parthenogenetic species (**A**–**D**) and two non-parthenogenetic species (**E**,**F**). Note similar morphology in all species with thick hilar (white double arrow) and thin distal (black double arrow) reservoir epithelium. BC: bursa copulatrix, R: reservoir, SG: spermatheca gland.

**Figure 4 insects-14-00195-f004:**
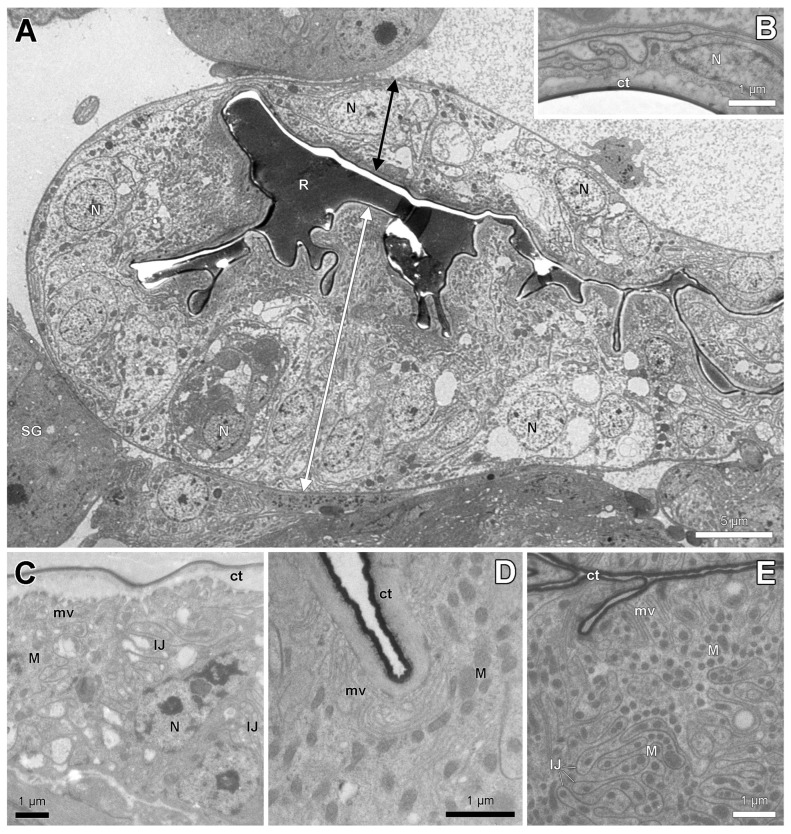
Electron micrographs of spermatheca reservoir of parthenogenetic queens. (**A**). General view of reservoir of *S. rogeri* with thick hilar (double white arrow) and thin distal (black double arrow) epithelium. (**B**). Thin distal epithelium with few cell organelles in *S. liukueiensis*. (**C**). Hilar epithelium in *S. liukueiensis* with microvilli (mv), mitochondria (M) and tortuous intercellular junctions (IJ). (**D**). Apical cytoplasm in *S. solifontis*. (**E**). Apical cytoplasm with abundant mitochondria in *S. rogeri*. ct: cuticle, N: nucleus, R: reservoir lumen, SG: spermatheca gland.

**Figure 5 insects-14-00195-f005:**
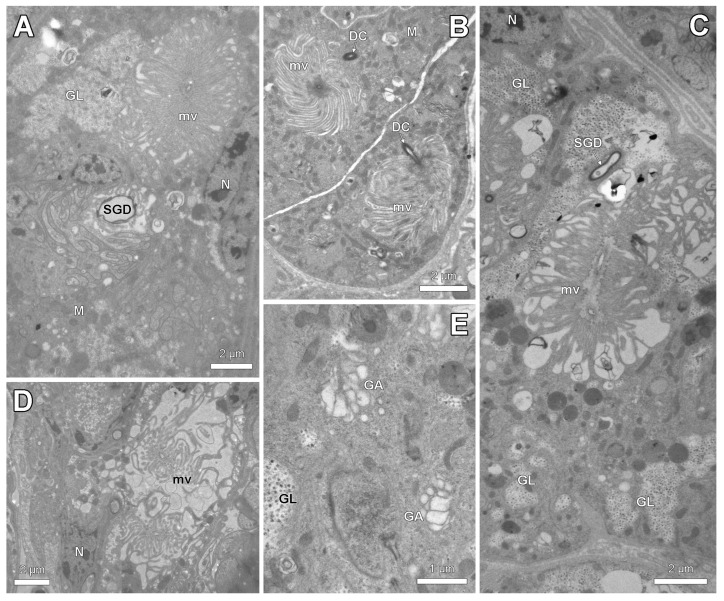
Electron micrographs of spermatheca gland of parthenogenetic queens. (**A**). General view showing spermatheca gland duct (SGD) and end apparatus with densely packed long microvilli (mv) in *S. liukueiensis*. Note accumulation of glycogen (GL). (**B**). Junction of duct cell (DC) with densely packed microvilli of end apparatus in *S. rogeri*. (**C**). Glycogen clusters and end apparatus with irregular microvilli in *S. solifontis*. (**D**). End apparatus with loosely arranged microvilli in *S. emmae*. (**E**). Detail of cytoplasm in *S. solifontis*, showing Golgi apparatus (GA) and glycogen. M: mitochondria, N: nucleus.

**Table 1 insects-14-00195-t001:** Species of which reproduction was studied during experimental period of 28 weeks (for *S. membranifera* 134 (*) and 202 weeks (**), respectively). Number of adult offspring during this period is listed. Bold values show total per species; grey background indicates thelytokous species.

Species (Collection Locality)	Colony	Experimental Period	Workers	Queens	Males
***S. emmae*** (Baihe Reservoir, Tainan City, Taiwan)	**7**		**367**	**18**	**0**
	I	10 December 2017–30 June 2018	65	0	0
	II	10 December 2017–30 June 2018	38	6	0
	III	10 December 2017–30 June 2018	53	3	0
	IV	10 December 2017–30 June 2018	60	6	0
	V	10 December 2017–30 June 2018	42	3	0
	VI	10 December 2017–30 June 2018	45	0	0
	VII	10 December 2017–30 June 2018	64	0	0
***S. lacunosa*** (Lugu Township, Nantou County, Taiwan)	**3**		**0**	**0**	**7**
	I	10 December 2017–30 June 2018	0	0	1
	II	10 December 2017–30 June 2018	0	0	2
	III	10 December 2017–30 June 2018	0	0	4
***S. liukueiensis*** (Jiji Township, Nantou County, Taiwan)	**5**		**225**	**12**	**0**
	I	10 December 2017–30 June 2018	54	1	0
	II	10 December 2017–30 June 2018	52	3	0
	III	10 December 2017–30 June 2018	42	4	0
	IV	10 December 2017–30 June 2018	37	3	0
	V	10 December 2017–30 June 2018	40	1	0
***S. membranifera*** (Huisun Forest, Nantou County, Taiwan)	**3**		**325**	**1 ***	**1 ****
	I	16 July 2017–1 June 2021	120	0	0
	II	16 July 2017–1 June 2021	130	0	0
	III	16 July 2017–1 June 2021	75	1 *	1 **
***S. minutula*** (Hengchun Township, Pingtung County, Taiwan)	**2**		**0**	**0**	**3**
	I	10 December 2017–30 June 2018	0	0	2
	II	10 December 2017–30 June 2018	0	0	1
***S. rogeri*** (Jiji Township, Nantou County, Taiwan)	**6**		**301**	**12**	**1**
	I	20 December 2017–30 June 2018	45	2	0
	II	20 December 2017–30 June 2018	81	3	0
	III	20 December 2017–30 June 2018	57	1	0
	IV	20 December 2017–30 June 2018	41	1	0
	V	20 December 2017–30 June 2018	40	4	0
	VI	20 December 2017–30 June 2018	37	1	1
***S. solifontis*** (Sun Moon Lake, Nantou County, Taiwan)	**3**		**153**	**7**	**0**
	I	5 July 2018–15 January 2019	38	2	0
	II	5 July 2018–15 January 2019	59	2	0
	III	5 July 2018–15 January 2019	56	3	0

**Table 2 insects-14-00195-t002:** Survey of species of which field-collected dealate queens were dissected to determine their insemination status (N = number of colonies). Grey indicates thelytokous species.

Species	Collection Locality and Date	N	Dissected Queens	Mated Queens	Virgin Queens
** *S. formosensis* **	Ren’ai Township, Nantou County, Taiwan (16 September 2022)	3	3	3	0
** *S. hispida* **	Yuchi Township, Nantou County, Taiwan (12 December 2016)	1	8	7	1
** *S. lacunosa* **	Lugu Township, Nantou County, Taiwan (20 October 2022)	2	2	2	0
** *S. liukueiensis* **	Jiji Township, Nantou County, Taiwan (20 October 2022)	3	9	0	9
** *S. minutula* **	Huisun Forest, Nantou County, Taiwan (11 December 2016)	1	2	2	0
** *S. rogeri* **	Jiji Township, Nantou County, Taiwan (16 September 2022)	2	11	0	11
** *S. solifontis* **	Huisun Forest, Nantou County, Taiwan (12 December 2016)	1	1	0	1
** *S. solifontis* **	Sun Moon Lake, Nantou County, Taiwan (13 December 2016)	7	21	0	21

## Data Availability

All microscopy data are available in the Zoological Institute, LSSE Lab, University of Leuven, Belgium.

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
