# Peer review of "Parthenogenetic Reproduction in Strumigenys Ants: An Update"

_insects, 2023, doi:10.3390/insects14020195_

Round 1

Reviewer 1 Report

In their manuscript titled " Parthenogenetic reproduction in Strumigenys ants: an update, authors discovered thelytokous parthenogenetic species and described their spermatheca morphology.

The research is crucial for future studies to reveal the evolutionary process of thelytokous parthenogenesis although they are evolutionarily dead-end. However, the crucial evidence indicating thelytokous parthenogenesis is not sufficient, and their explanations about the experiments are not clear.

I have comments as follows.

Table 1: It is difficult for me to understand many points. 

Please explain what the asterisks indicate in legend. Maybe they indicate experimental period or emergence date of queens and males during experimental period? Why are the asterisks absent in colony III of S. membranifera?

The experimental period, the authors described 28 weeks in materials and methods and 196 days in Table 1 legend. Please integrate them.

In S. membranifera, I think the experimental period is not 28 weeks, so explain it in materials and methods.

I assume that S. rogeri colony I-VI are F0 and VII-XI are F1 generations from the experimental period. Are the other colonies in other species F0 generations? Is it correct? The authors should clarify the explanation in materials and methods or in table 1.

In line 77 in method section, the authors explained “six experimental groups were created, each consisting of one alate queens and 15 workers”. Which are six experimental groups in table 1? For instance, there are 7 colonies in S. emmae and 5 colonies in S. liukueiensis, in which they are not six colonies. Are they not the experimental group? Or, do the Roman numbered codes in table 1 indicate mother colony code, and did the author create six experimental groups from the mother colony, and are numbers of newly emerged workers/queens/males in table 1 the sum of the individuals from six experimental groups? Clear them.

Did the authors confirm whether the one mother alate queen from all experimental groups was virgin by dissecting and checking the spermatheca after the experimental period? If so, describe it as mentioned in Lee et al (2018). This is essential evidence for the decision of thelytokous parthenogenesis. In histological analyses, did the authors use the mother alate queens in table 1 after the experimental period? From the sections of abdomens, the sperm presence or absence in the spermathecal reservoir can be checked (as described in line 133), however, I think it is difficult that all specimens are successful for sectioning, so that I recommend that the authors mention how many queens do not have spermatozoa in the spermatheca.

Table 2:

The authors should provide more colony information. Were they dealated queens? Did they have daughter workers? What was their colony composition? This is also crucial information to indicate thelytokous parthenogenesis. Moreover, were the queens originate from one mother nest per one collection site? How many colonies did you observe in each species? Were these colonies identical to the mother colonies used in table 1?

When were they collected? I think it is important to argue collection seasons and alate production seasons in Strumigenys species investigated here because it may be possible that daughter queens which defect mating and discard wings still stay in the mother nest soon after mating season.

Clear them in materials and methods, or results.

In discussion, the authors should explain morphologies of degenerated spermatheca referring Gobin et al. (2006). It may be helpful for readers to understand which cell features are essential for functional spermatheca.

I recommend legend and image of Fig. 1 together with in the same page (Maybe, the editor can make the image a little bit small. This is not strange as a design.). The authors should tell the editor.

Others:

Line 32: Scientific names should be written in italic.

Line119-123 and 152-155: The terms, “unpaired cuticle-lined oviduct”, “lateral oviduct” and “paired oviduct” are confusing because different words were used in result and Fig. 1 legend. I think “lateral oviduct” is same as “paired oviduct”, and “unpaired cuticle-lined oviduct” means common oviduct (Maybe, the term “common oviduct” is more frequently used in anatomy of insects.). If the authors have reasons to use the different terms (“lateral oviduct” vs “paired oviduct”), explain them in result. If not so, integrate them.

In genus Strumigenys, there are many thelytokous parthenogenetic species than other ant genera. If the authors have some idea why Strumigenys species are likely to evolve the reproductive mode, please explain it in discussion.

And, if phylogenic relationships of Strumigenys species are available, assign the thelytokous parthenogenetic species to the phylogeny. This leads understanding of the evolutionary process of thelytokous parthenogenesis among Strumigenys

Reviewer 3 Report

Thanks for this submission - which I think will make a very good addition to the literature on thelytoky in ants (including the discovery of several more thelytokous species).  Overall, I was satisfied with your paper and found the histological and electron micrograph images clear).  Aside from some minor editing of the English (a few grammatical glitches), I recommend your paper for publication.

Reviewer 4 Report

The work by Wang st al. on the parthenogenetic reproduction in Strumigenys ants is a nice work, well written and accompanied by clear and exhaustive iconography. The research describes, the particular finding present in three ant species, belonging to the genus Strumigenys characterized by telitoky, which means that queen females  of these species do not need to receive sperm from males at mating and can produce female offspring. Quite interesting, these telytokous females still have a functional spermatheca, provided with a spermatheca gland, which can store sperm in case of future mating with males.

The text of the work is accompanied by very useful and good semithin sections which illustrate the presence of the whole female genital apparatus, with the spermatheca perfectly preserved. The micrographs of the different organs are of good quality and support the text description. 

This reviewer wish to express his compliments to the Authors for the interesting work they were able to produce. 

IIn the opinion of this reviewer the work is recommended for publication in buy Insects

Reviewer 5 Report

This manuscript is excellently written, clear and concise. The experimental design is logically sound and methods are elegant and detailed so as to make this work replicable.

The work on parthenogenesis is an important and timely in helping to determine potential problematic tramp species. I like that the authors make this connection. As an aside, I have not collected nor know of a collection of male S. eggersi - which is now one of if not the most commonly collected ant in natural areas of Florida now.

As far as lineages and loss of genetics, the system of V. emeryi gives some hints. Gynandromorphy is likely due to the complete absence of sexual reproduction. There is a mention of the advantage of keeping potential sexual reproduction, the system of V. emeryi might give some more citation support to this statement.

I had no issues with this manuscript, but would suggest authors check antweb, antmaps, and antwiki for other collections of males. I am providing a list of males I have examined or are available at AntWeb.

SpecimenCode Notes Subfamily Genus Species DeterminedBy OwnedBy LifeStageSex Medium CollectionCode CollectedBy Method DateCollectedStart DateCollectedEnd Habitat LocalityCode LocalityName Adm1 Adm2 Country NativeBiogeographicRegion Elevation ElevationMaxError LocLatitude LocLongitude
mem087169 confirmed in Booher et al. 2021 Myrmicinae Strumigenys membranifera D. Booher D. Booher 1m pin ANTC34054 T.L. Schiefer, D.M. Burge malaise 7/6/02 7/13/02 deciduous forest White Sand 40 White Sand MS Pearl River Co United States Nearctic 40   30.793889 -89.659444
casent0160797   myrmicinae strumigenys rogeri B. Fisher CASC, San Francisco, CA, USA 1m pin BLF23134 B.L.Fisher et al. Malaise trap 1/20/10 1/28/10 forest Silhouette 520 Silhouette Island, above Jardin Marron on crest to Mont Plaisir and Pot à Eau Seychelles Malagasy 520   -4.4867 55.2341
casent0137483   myrmicinae strumigenys rogeri B. Fisher CASC, San Francisco, CA, USA 1m pin BLF19767 B.L.Fisher et al. 9 MaxiWinks, mixed samples 3/15/08 3/17/08 montane rainforest Grille 995 Grillé Grande Comore Comoros Malagasy 995   -11.47578 43.34669
casent0133445 S. emmae was the only Strumigenys collected in this sample. There were many workers, male is likely S. emmae but with no  myrmicinae strumigenys emmae B. Fisher CASC, San Francisco, CA, USA 1w pin BLF19060 B.L.Fisher et al. 9 MaxiWinks, mixed samples 12/2/07 12/4/07 rainforest Dapani 135 Dapani     Mayotte Malagasy 135   -12.96279 45.15037
CASENT0133418 information here https://www.antweb.org/specimen.do?name=casent0133418                                  
